**www.cambridge.org/ext**

IUCN Red List; biodiversity threats; body size; marine ray-finned fishes; extinction risk

**Corresponding author:**
Trevor M. Bak;
Email: tbak@hawaii.edu

# A global ecological signal of extinction risk in marine ray-finned fishes (class Actinopterygii)

Trevor M. Bak[1] ⬤, Richard J. Camp[2] ⬤, Noel A. Heim[3] ⬤, Douglas J. McCauley[4] ⬤, Jonathan L. Payne[5] ⬤ and Matthew L. Knope[6] ⬤

[1]Tropical Conservation Biology and Environmental Science Graduate Program, University of Hawaiʻi at Hilo, Hilo, HI, USA; [2]U.S. Geological Survey, Pacific Island Ecosystems Research Center, Hawaiʻi National Park, HI, USA; [3]Department of Earth & Ocean Sciences, Tufts University, Medford, MA, USA; [4]Department of Ecology, Evolution, and Marine Biology and Marine Science Institute, University of California, Santa Barbara, CA, USA; [5]Department of Geological Sciences, Stanford University, Stanford, CA, USA and [6]Department of Biology, University of Hawaiʻi at Hilo, Hilo, HI, USA

## Abstract

Many marine fish species are experiencing population declines, but their extinction risk profiles are largely understudied in comparison to their terrestrial vertebrate counterparts. Selective extinction of marine fish species may result in rapid alteration of the structure and function of ocean ecosystems. In this study, we compiled an ecological trait dataset for 8,185 species of marine ray-finned fishes (class Actinopterygii) from FishBase and used phylogenetic generalized linear models to examine which ecological traits are associated with increased extinction risk, based on the International Union for the Conservation of Nature Red List. We also assessed which threat types may be driving these species toward greater extinction risk and whether threatened species face a greater average number of threat types than non-threatened species. We found that larger body size and/or fishes with life histories involving movement between marine, brackish, and freshwater environments are associated with elevated extinction risk. Commercial harvesting threatens the greatest number of species, followed by pollution, development, and then climate change. We also found that threatened species, on average, face a significantly greater number of threat types than non-threatened species. These results can be used by resource managers to help address the heightened extinction risk patterns we found.

## Impact statement

Earth is experiencing a sustained decline in biodiversity, and broad global-level perspectives can point to what species are at the highest risk of extinction, which can be used in prioritizing conservation efforts. Additionally, a better understanding of the extinction risk of lesser-studied species could help prevent biodiversity loss across all taxa. In this study, we address the extinction risk of marine ray-finned fishes using a global dataset and explore what ecological traits are associated with extinction risk, what threat types are contributing to extinction risk, and whether threatened species face a greater number of threats than non-threatened species. We find that larger species and those that move between marine, brackish, and freshwater are at higher extinction risk. We also find support that commercial harvesting is likely pushing larger species toward extinction, and that threatened species are impacted by a greater number of threat types than non-threatened species. Cumulatively, our results highlight global extinction risk patterns among marine ray-finned fishes; reducing fishing pressure on larger species and protecting those that move between marine, brackish, and freshwater from pollution and development are management strategies that can help protect these threatened species from extinction.

## Introduction

The current rate of extinction in many taxonomic groups is so high that modern biodiversity loss is often referred to as heralding the sixth mass extinction (Barnosky et al., 2011) in reference to the "Big Five" mass extinctions first identified in the marine animal fossil record (Raup and Sepkoski, 1982). Indeed, the current rate of extinction has been estimated to be as much as a thousand times higher than the background rate of extinction (Ceballos et al., 2010; De Vos et al., 2015). However, extinctions of relatively few marine species have been documented (Hutchings and Reynolds, 2004; Monte-Luna et al., 2007, 2023). Widespread human impact has occurred more recently in the oceans than on land (McCauley et al., 2015), but the low number of recorded marine extinctions may also reflect, to some extent, an undercounting bias due to more limited research attention (Webb and Mindel, 2015). Despite the few documented extinctions to date, many

species are in decline, and threat levels are high in some taxonomic groups (Reynolds et al., 2005; McCauley et al., 2015), suggesting that extinction rates may become higher in the near future (Monte-Luna et al., 2007; Ceballos et al., 2017). Marine fishes, in particular, are under intense pressure from overharvesting, in addition to a variety of other local and global threats (Cheung et al., 2013; McCauley et al., 2015; Pauly and Zeller, 2016; IPBES, 2019), with some fish populations showing as much as 90% reduction in their historical geographic ranges (McCauley et al., 2015; Newsome et al., 2020).

Although the rate and magnitude of marine fish population declines have been well studied (Reynolds et al., 2005; McCauley et al., 2015), the taxonomic and ecological distributions of these declines and their associated threats remain understudied. If the rate of marine fish extinction increases, understanding extinction selectivity, or the traits associated with extinction, will be key to understanding how biodiversity loss may potentially reshape marine ecosystems (Bush et al., 2020). Determining whether extinction will be selective is especially critical because nonrandom extinction tends to lead to greater loss of evolutionary history than random extinction (Purvis et al., 2000a; Vamosi and Wilson, 2008), potentially requiring many millions of years for recovery (Davis et al., 2018). Pinpointing the ecological characteristics of fishes at higher extinction risk can help identify species that could be prioritized for conservation management and the strategies that could be most effective, helping to direct limited resources.

Marine ray-finned fishes (class Actinopterygii) are the most speciose clade of marine vertebrates, central to global marine ecosystem function and services, and are of high economic and nutritional importance to human communities (Helfman and Naiman, 2009). Yet, their extinction risk profiles are not as well understood as terrestrial vertebrates (Atwood et al., 2020; Munstermann et al., 2022) or cartilaginous fishes (Dulvy et al., 2014; McCauley et al., 2015). To assess extinction risk selectivity among marine ray-finned fishes, we used the International Union for Conservation of Nature's (IUCN) Red List (IUCN, 2022). Previous studies have examined the selectivity of modern extinction threat (Purvis et al., 2000b; Reynolds et al., 2005; Olden et al., 2007; Jager et al., 2008; Payne et al., 2016; Ripple et al., 2017; Munstermann et al., 2022), but prior studies have not focused on ray-finned fishes at the species level with both the breadth of ecological traits and the number of species examined here, as these expanded datasets have only recently become available (Froese and Pauly, 2022; IUCN, 2022). We also capitalized on recently available, large-scale, species-level phylogenetic hypotheses for marine ray-finned fishes (Rabosky et al., 2018) to address possible phylogenetic non-independence of species ecological traits. Few prior studies of extinction risk in marine fishes have incorporated phylogenetic hypotheses into their analyses, and this lack of phylogenetic data could yield misleading results if the predictor and response variables are strongly phylogenetically nested (Purvis et al., 2000b; Harnik et al., 2012; Finnegan et al., 2015). We also analyzed the association of our ecological trait variables and the IUCN Red List threat types to explore what threats are most responsible for pushing these fishes toward greater extinction risk.

In this study, we explored how a suite of biologically important traits may be associated with extinction risk and the most common extinction threat types. To do so, we addressed the following three questions: (1) What is the association between ecological traits and extinction risk? (2) Do threatened species have a greater number of threats assigned by the IUCN Red List on average than non-threatened species? (3) Is there an association between ecological traits and the threat types that species face?

## Methods

### Datasets

We created a master dataset by joining together three separate datasets: the IUCN Red List (IUCN, 2022), World Register of Marine Species (WoRMs) (Horton et al., 2022), and FishBase.org (FishBase) (Froese and Pauly, 2022). All analyses were conducted in R 4.1.3 (R Core Team, 2022), and the data and code are available at https://github.com/TrevorBak/Marine-Ray-Finned-Fishes-Extinction-Risk. We downloaded all species with an IUCN Red List assignment in May 2022 and then filtered to include only the ray-finned fishes (class Actinopterygii). Separately we downloaded in May 2022 all species in WoRMs for kingdom Animalia. We inner-joined both datasets for the purpose of filtering out nonmarine Actinopterygiian species from the IUCN dataset. A separate dataset was created with the ecological predictor variables by extracting data from FishBase for all fish species in the database in January 2021. We then inner-joined the FishBase dataset with the combined IUCN–WoRMs dataset, resulting in a dataset of 9,966 marine ray-finned fish species that have both an IUCN Red List assignment and ecological trait data from FishBase. Out of the 9,966 species, 1,769 were data-deficient and removed, for a final dataset of 8,155 species.

### Predictor variables

The predictor variables from FishBase that we examined were body size, minimum population doubling time, habitat tiering, euryhaline status, and trophic level. We considered using a range of other variables from FishBase but decided not to include due to lack of comprehensive data (e.g., max weight), concerns about circularity (e.g., vulnerability index), similarity to other variables we were already evaluating (e.g., generation time), or being unable to infer cause-versus-effect (e.g., for price; is price driving extinction risk or reflective of higher extinction risk?). Body size was measured as total length (millimeters) at adult stage and then $\log_{10}$-transformed for analysis. In addition to testing how body size was associated with extinction risk, we also plotted a percent histogram exploring what percentage of species were threatened as a function of body size. Minimum population doubling time was based on Froese et al. (2017) and converted into a categorical variable by FishBase, consisting of <1.4 years, 1.4–4.4 years, 4.5–15 years, or >15 years. For habitat tiering, we grouped a range of tiering classifications (e.g., demersal, pelagic-oceanic, bathypelagic) into the coarse binomial categories of benthic or pelagic to increase sample size and statistical power. We assigned euryhaline status by taking the binary categorizations created by FishBase, indicating whether or not a species was marine, brackish, or freshwater, and using these categorizations to classify species as "marine only," "marine and brackish," or "marine, brackish, and freshwater." Trophic position was defined as mean trophic position of the food items the species eats (Froese and Pauly, 2000).

### Extinction risk

The IUCN Red List categorizes extinction risk into seven ordered categories (from least to most threatened): least concern (LC), near threatened (NT), vulnerable (VU), endangered (EN), critically endangered (CR), extinct in the wild (EW), or extinct (EX). Species lacking sufficient data to make a categorization are classified as data-deficient. Following the IUCN convention of considering species in the vulnerable, endangered, and critically

endangered categories as being 'threatened' and species in the least concern and near-threatened categories as being 'non-threatened' (IUCN, 2022), extinction risk status was collapsed to this binomial designation to increase statistical power. We did not include extinct in the wild ($N = 1$) or extinct ($N = 4$) species due to these species no longer being a part of modern ecosystems. Given the small number of such species, this exclusion is unlikely to affect the results, in line with previous analyses (Olden et al., 2007; Atwood et al., 2020). To investigate how the cutoff for the binary classification schema (threatened or non-threatened) may impact our results, we performed a separate sensitivity analysis that was designed to test how robust our results are to the binary categorization. In the sensitivity analysis, vulnerable species were classified as 'non-threatened' to explore if extinction risk patterns change when only the most vulnerable species (EN and CR) are included in the 'threatened' category.

## Generalized linear models

We used generalized linear models to test for possible associations between predictor variables with both extinction risk and threat types as response variables. We performed a logistic regression with binomial distribution using the logit link function. Our response variables were binomial (threatened or non-threatened, threat assigned or not assigned), while the predictor variables were a mix of quantitative, ordered, and categorical variables. For categorical variables, all results are relative to a reference level of the variable. For minimum population doubling time, we set the reference category to the fastest doubling interval (<1.4 years) to test if longer population doubling times influenced extinction risk. For euryhaline status, we set "marine only" as the reference to test if living in part brackish or freshwater influenced extinction risk. For tiering, we set pelagic as the reference to test if benthic species were at significantly different extinction risk. We kept the reference categories the same for the threat type analyses for consistency.

## Phylogenetic generalized linear models

We incorporated phylogenetic hypotheses into our generalized linear models by using phylogenetic generalized linear models (phyloGLMs) to analyze associations among our five ecological traits and both extinction risk status and threat types, while accounting for statistical non-independence due to shared evolutionary history (Felsenstein, 1985). We downloaded species-level molecular phylogenetic trees for Actinopterygii from the data supplement in Rabosky et al. (2018), and completed all analyses using the Phylolm package (Tung Ho & Ané, 2014). Trees were initially built by Rabosky et al. (2018) using 24 nuclear and mitochondrial loci from five sources (see Rabosky et al., 2018; Supplementary Table S1 for details) in a Bayesian phylogenetic framework (see Rabosky et al., 2018; Supplementary Table S2 and S5 for details), with species missing molecular data added according to known taxonomic relationships. Rabosky et al. (2018) repeated this estimation 100 times to generate 100 trees and produce a distribution of fully sampled Actinopterygii.

## AIC model selection

We used Akaike information criterion (AIC) model selection (Burnham and Anderson, 2004) to determine which variable(s) best predicted extinction risk. We developed a set of 10 candidate models based on biological hypotheses and published literature of how our variables may be associated with extinction risk and chose the model with the lowest AIC value to make inference. The 10 candidate models included a null model, five univariate models (a single variable per model), three multivariate additive models, and a fully parameterized additive model (Supplementary Table S1).

## Analysis of critically endangered species

We also explored the association of predictor variables for the species in the critically endangered (CR) Red List status to highlight the extinction profile of species on the brink of extinction. For this we created a subset of data that included only the critically endangered species. We then tested for possible association with variables from our best model as determined by AIC selection. For body size (total length), we utilized Kruskal–Wallis and Dunn multiple comparison tests. For euryhaline status, we used a Chi-square test and a comparison of standardized residuals from the Chi-square test to determine which observations differed the most from the expected proportion of equal likelihood for each euryhaline category.

## Threat types

We downloaded extinction threat types from the IUCN Red List in May 2022. The 12 primary IUCN RedList threat types are (1) Development, (2) Aquaculture and Agriculture, (3) Energy Production and Mining, (4) Transportation, (5) Harvesting, (6) Human Disturbance, (7) Natural System Modification, (8) Invasive Species and Disease, (9) Pollution, (10) Geological Events, (11) Climate Change, and (12) Other. A total of 3,014 of the 8,185 species in our final dataset had one or more threats assigned to them. We determined what the most common threats were across all of these species using the IUCN assignments, and we tested if threatened species had a greater average number of assigned threats compared to non-threatened species using Wilcoxon rank-sum tests. Lastly, we explored how each predictor variable was associated with each of the four most common threat types using PhyloGLMs. We set each of the top four threat types as the response variable (binomial, threat assigned or not) using four different phyloGLM models (e.g., model1 = phyloglm(Harvesting ~ Log_BodySize + Euryhaline_Status)).

## Sample sizes

The sample size ($N$) varied among our three main analyses depending on data availability. For the first analysis examining predictor variables and extinction risk, we limited the dataset to complete cases ($N = 1,766$). That is, species for which data were present across all five ecological variables. Sample size by extinction risk category was: least concern ($N = 1,632$), near threatened ($N = 35$), vulnerable ($N = 52$), endangered ($N = 25$), and critically endangered ($N = 22$). Collapsed to binary, this resulted in 1,667 species classified as non-threatened and 99 as threatened. The sensitivity analysis moved the vulnerable species from being threatened to non-threatened, resulting in a new sample size of 1,719 species classified as non-threatened and 47 as threatened. For the second analysis, comparing the number of threats assigned between threatened and non-threatened species, the sample is the complete dataset of 8,155 species. For the third analysis, exploring the association of predictor variables from the best model (as chosen by AIC; see below), sample size was 4,433 representing the species that had data for both euryhaline status and body size.

## Results

### AIC model selection

An AIC model selection process determined that the best-fit model of extinction risk status included only body size and euryhaline status as predictors (Supplementary Table S1). AIC weight for the body size and euryhaline status model was 68% (Supplementary Table S1). The next best-fit AIC model, euryhaline status alone, was within 2 AIC units of our chosen model and therefore statistically an equally good fit (Burnham and Anderson, 2004). Because the two models were statistically indistinguishable, we used the body size and euryhaline status to more fully examine the role of both body size and euryhaline status on extinction risk.

### Predictor variables and extinction risk

We found statistically significant relationships for body size and for euryhaline status with extinction risk, with larger fishes and fishes that come into contact with marine, brackish, and freshwater during their life history at higher risk (Figure 1). The distribution of coefficients for each predictor variable displays how each variable is associated with threatened status across the 100 phylogenetic trees. Coefficient distributions that fall completely above or completely below zero indicated a statistically significant association (alpha = 0.01; Munstermann et al., 2022). Estimates are presented as log odds, which is the log of the odds ratio of being threatened by extinction. The percentage of species threatened as a function of body size showed both larger and (to a much lesser degree) smaller fishes having a higher percentage listed as threatened relative to intermediate-sized fishes (Figure 2). However, overall, threatened fishes were on average larger (*t*-test: non-threatened mean 353.20 (mm), threatened mean 691.93 (mm), $t = -9.9$, df = 341.23,

$p < 0.001$). Moving vulnerable species from threatened to non-threatened affected the results (Figure 1), with "marine and brackish" also becoming significantly associated with extinction risk. In addition, for both "marine, brackish, and freshwater" and body size, the coefficients became larger when vulnerable species were assigned to the non-threatened group.

### Critically endangered species

Critically endangered species were on average larger than species in the other IUCN Red List risk categories (Kruskal–Wallis test, $k = 40.8$, $df = 3$, $p < 0.001$; Figure 3a and Supplementary Table S3) (mean CR 1691.87 mm, mean EN 624.02 mm, mean VU 621.90 mm, mean non-threatened 358.96 mm). Critically endangered species also had a greater proportion of "marine, brackish, and freshwater" fishes than other risk categories ($\chi^2$ test, $x = 260.68$, $df = 6$, $p < 0.001$; Figure 3b and Supplementary Table S3). A comparison of standardized residuals (Supplementary Table S4) indicated that the critically endangered "marine, brackish, and freshwater fishes" have the greatest deviance from the expected value (standardized residual 12.44).

### Threat types

The most common threat type across all species was harvesting (1,787 species), followed by pollution (718 species), development (572 species), and then climate change (516 species) (Figure 4a). Threatened species had a greater median number of total threat types assigned than non-threatened species (median threatened 2, median non-threatened 0) (Wilcoxon sign-rank test, $w = 405,929$, denominator $df = 8,113$, $p < 0.001$; Figure 4b).

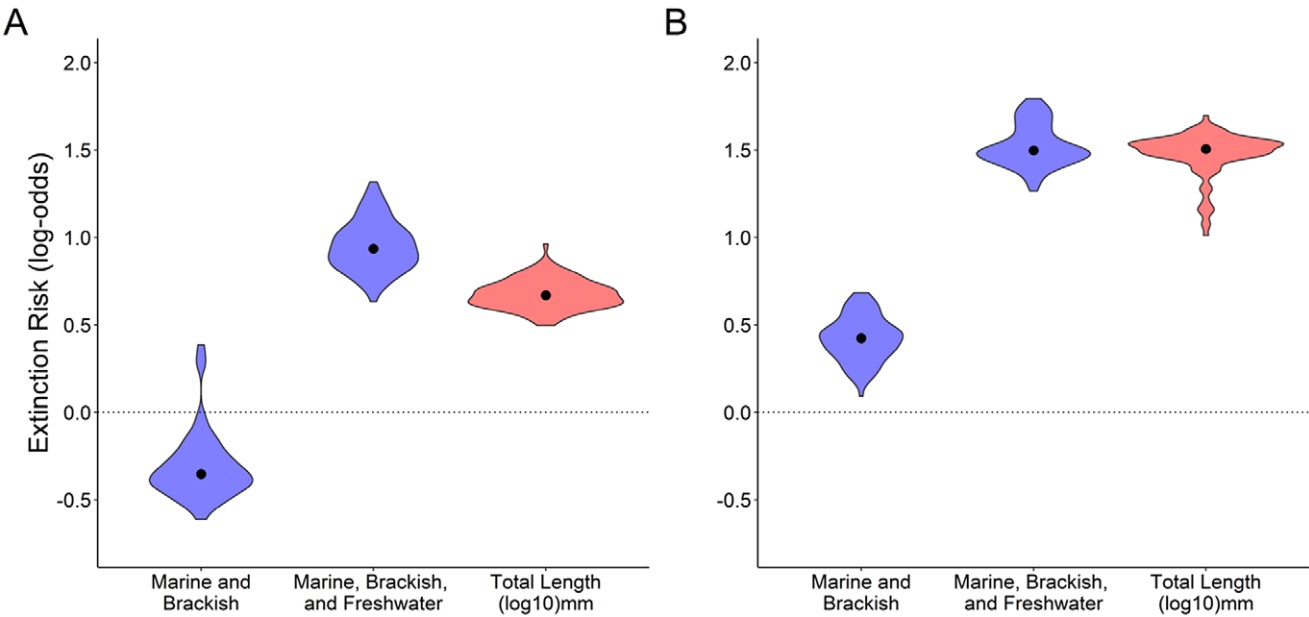

**Figure 1.** (a) Results from the phyloGLM analysis across all 100 phylogenetic trees testing for an association between both euryhaline status and body size (total length in $\log_{10}$ mm) and extinction risk (log-odds), with vulnerable species classified as threatened. The zero line represents no difference from the reference category (marine only for euryhaline status); coefficients falling completely above or below zero are statistically significant at alpha = 0.01 level (Munstermann et al., 2022). The violin plot shows the spread of coefficient values with width indicating the number of coefficient values, and median values indicated by black dots. (b) Results from the phyloGLM analysis using the same methodology, but with vulnerable species classified as non-threatened.

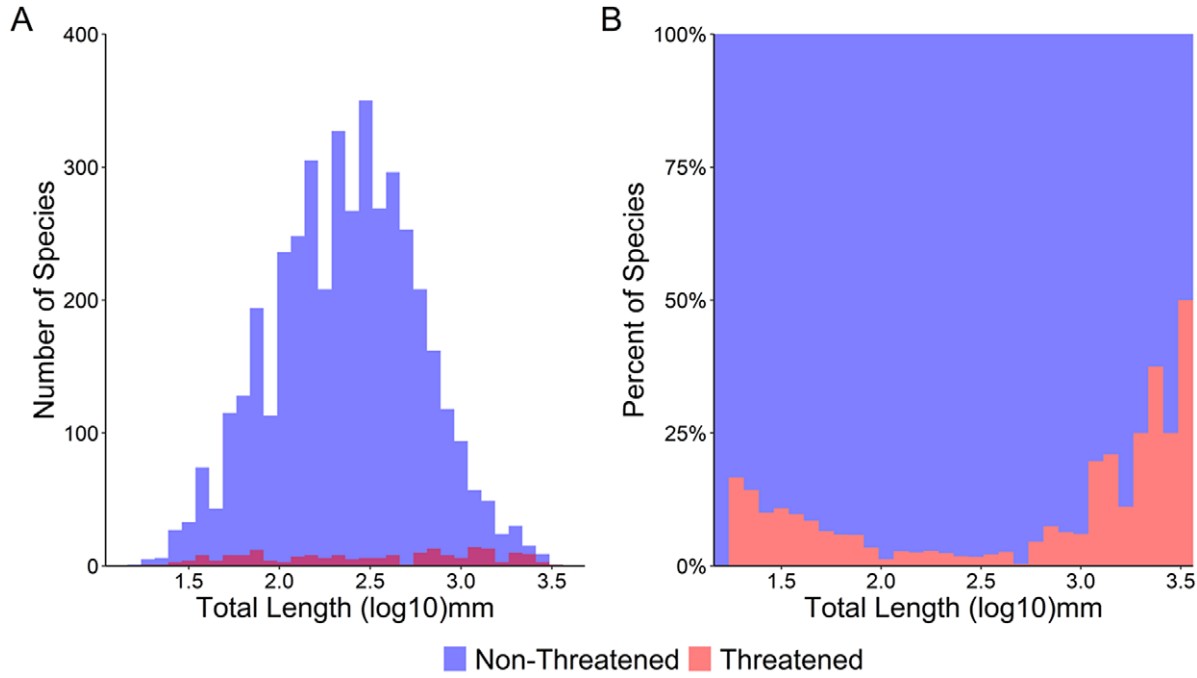

**Figure 2.** (a) Number of species threatened versus non-threatened. (b) Percentage of species threatened versus non-threatened as a function of body size (total length in $\log_{10}$ mm).

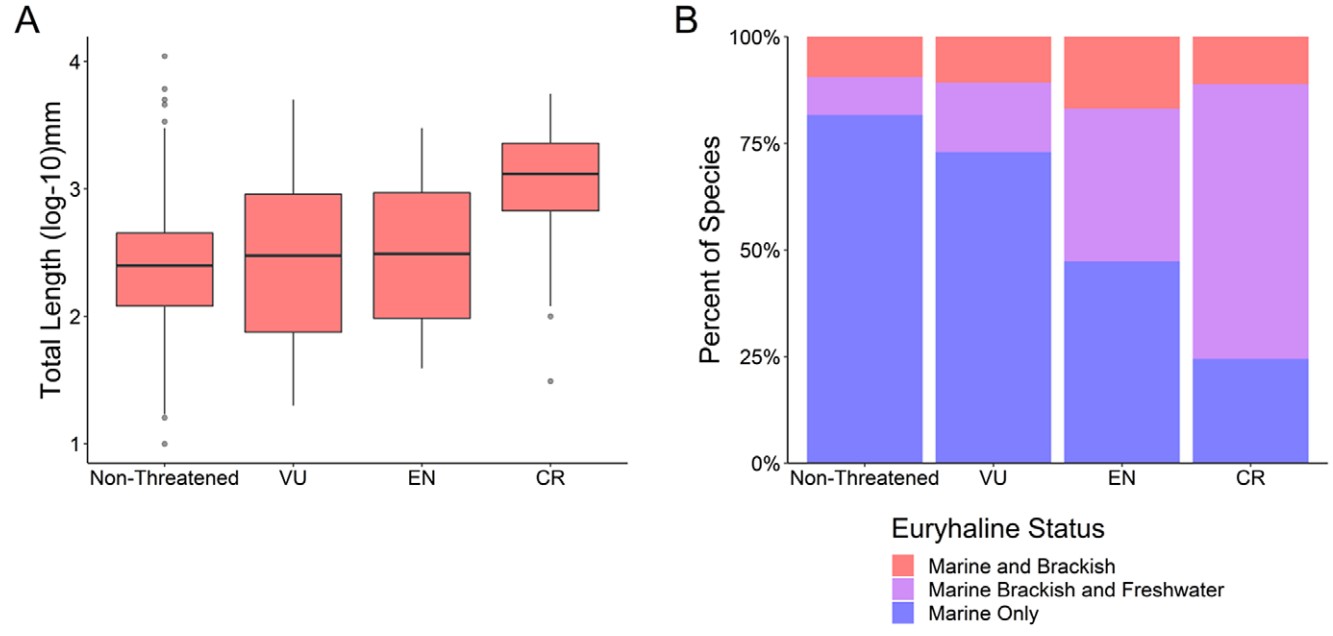

**Figure 3.** (a) Body size (total length $\log_{10}$ mm) as a function of IUCN Red List risk categories: Non-threatened (least concern and near threatened), vulnerable (VU), endangered (EN), critically endangered (CR). (b) Euryhaline status as a function of the same IUCN Red List risk categories.

### Predictor variables and the top four threat types

We tested the association between the ecological variables included in the best model (both body size and euryhaline status) and the top four threat types: harvesting, pollution, development, and climate change (Supplementary Table S6). We chose to test only the top four threats due to too few species being assigned to the other eight Red List threat types for robust inference. For body size, larger fishes were significantly more likely to be threatened by harvesting, while smaller fishes were significantly more likely to be threatened by pollution ($p < 0.01$). Development and climate change did not have a significant association with body size ($p > 0.01$; Figure 5). For euryhaline status, "marine and brackish" and "marine, brackish, and freshwater" had a significant positive association with harvesting, pollution, and development ($p < 0.01$, respectively), but no significant association with climate change ($p > 0.01$) (Figure 5).

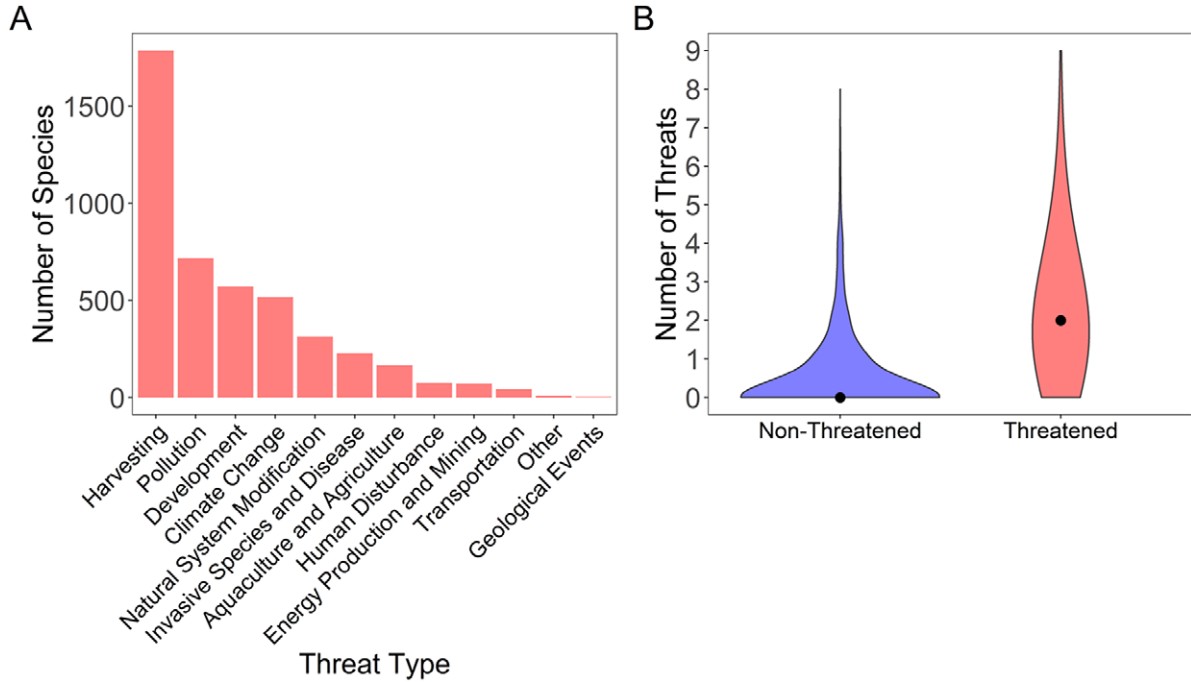

**Figure 4.** (a) Number of species threatened by each threat type, across all species in the dataset with IUCN Red List threat types assigned. (b) Distribution of the number of threats for non-threatened and threatened species. Black dots represent median values.

**Figure 5.** Results for phyloGLM analysis across all 100 phylogenetic trees testing for association of both total length (log$_{10}$ mm) and euryhaline status with each of the top four threat types: (a) harvesting, (b) pollution, (c) development, and (d) climate change. The zero line represents no difference from the reference category (marine only) for euryhaline status; coefficients falling completely above or below zero are statistically significant at alpha = 0.01 level (Munstermann et al., 2022). The violin plot shows the spread of coefficient values with width indicating the number of coefficient values, and median values represented by black dots.

## Discussion

This study provides novel insights into the well-documented declines in marine ray-finned fishes by examining ecological extinction threat profiles in a phylogenetically explicit framework. We found larger fishes and/or fishes that move between marine, brackish, and freshwater environments to be at the greatest extinction risk (Figure 1). We found that harvesting is likely driving the heightened extinction risk of larger fish species, while pollution may be driving the increase in extinction risk for smaller fish species (Figure 5). Harvesting, pollution, and development may also be driving the heightened extinction risk of fishes that come into contact with freshwater and/or brackish water, relative to marine-only species (Figure 5). Together, our results illuminate which types of marine ray-finned fish species are at greater extinction risk and which threat types could be prioritized to potentially prevent these extinctions.

### Traits associated with extinction risk

Humans have been selectively harvesting larger-sized fish species for at least the past 50,000 years (Jackson et al., 2001), and this trend continues. We found that larger fish species are at greater extinction risk (Figure 1) while also detecting a less-pronounced rise in extinction risk for the smallest species (Figure 2), in line with the results of previous studies (Reynolds et al., 2005; Olden et al., 2007; Pinsky et al., 2011; Ripple et al., 2017). This size bias of extinction risk for larger species is especially pronounced for the critically endangered fishes (Figure 3a). This higher extinction risk among larger species may be unlike that of previous mass extinction events; in the "big five" mass extinctions (Raup and Sepkoski, 1982), either no size bias was detected or smaller marine genera had a higher likelihood of extinction (Payne et al., 2016; Puttick et al., 2017). Friedman et al. (2009) have detected a weak bias against larger fish genera for the Cretaceous–Paleogene (K–Pg) extinction event; however, this size bias did not remain statistically significant after controlling for phylogeny. Therefore, it appears that this size selectivity in marine ray-finned fishes may be a unique signal of modern extinction risk compared to previous extinction patterns, which has also been observed for marine vertebrates and mollusks when analyzed collectively (Payne et al., 2016).

We found that fishes that come into contact with brackish or freshwater as part of their life histories were at elevated extinction risk relative to marine-only fishes, in line with previous research that has found diadromous fishes are at a heightened extinction risk (Grant et al., 2019). While we do not evaluate diadromous migratory patterns directly, the decline of diadromous fishes is likely driving the pattern in our data of "marine, brackish, and freshwater" species being at higher risk. "Marine, brackish, and freshwater" fish species being at a higher risk than "marine only" and "marine and brackish" species indicates that these fishes are likely subjected to additional stressors not faced by fishes that do not enter freshwater during their lives (Figure 1). "Marine, brackish, and freshwater" fishes also make up over half the species in this study that are critically endangered (Figure 3). Our findings are in alignment with the pronounced declines in diadromous fish populations in the past century from overfishing, with some species experiencing declines of over 90% (Limburg and Waldman, 2009). Atlantic salmon (*Salmo salar*) catches have declined to near zero (Chaput, 2012), while Chinook salmon (*Oncorhynchus tshawytscha*) runs in Oregon (USA) are down to 11–19% of their estimated historical level (Meengs and Lackey, 2005). A genetic analysis of Pacific salmon (*Oncorhynchus* spp.) indicates that the century-long decline of these species may be greater than previously estimated (Price et al., 2019).

We did not find strong AIC support for trophic level or minimum population doubling time, suggesting these variables had weak explanatory power for explaining extinction risk. This raises the question of whether uncertainty in the models themselves used to generate these trait values contributed to the lack of explanatory power. For minimum population doubling time in particular, the intrinsic population growth rate ($r$) can be difficult to estimate for some fishes (Fishbase Manual, 2000), potentially leading to inaccurate population doubling time estimates. Minimum population doubling time is based on the Monte Carlo production model of Froese et al. (2017), and most population doubling times are derived from model predictions. In simulations, the model gave good predictions of parameters $r$ and $k$, and maximum sustainable yield and the Monte Carlo-based predictions were not significantly different in 76% of 128 real stocks with full stock assessments when evaluated against Bayesian Schaefer production model estimates (Froese et al., 2017), overall giving us confidence in the model. Trophic level is estimated from the Ecopath model (Pauly et al., 2000), and uncertainty in the estimates of trophic level may have contributed to low AIC support. However, trophic levels recorded in FishBase have been found to be accurate when compared to estimates using other methodologies such as stable isotope analysis (Mancinelli et al., 2013). Our aim in conducting a global analysis with the widest possible set of species does necessitate us taking a "least common denominator" approach, that is, using the dataset with the largest coverage which FishBase provides. One weakness of a least common denominator approach is a lack of access to the underlying data to perform analyses directly, but this approach allows us to evaluate fishes on an unprecedented scale, and the relatively high concurrence of the predicted parameters for simulations and real stocks and cross-validation of trophic level estimates lends confidence to the models themselves.

### Sensitivity analysis

Classifying vulnerable species as non-threatened increased coefficient values for body size, "marine and brackish," and "marine, brackish and freshwater," compared to the model with vulnerable species classified as threatened (Figure 1). The signal of fishes that move between marine, brackish, and freshwater environments and/or larger fishes being at greater risk is especially strong for species classified as being at the highest extinction risk (endangered or critically endangered; Figure 1). These results suggest that the general signal in the data (i.e., body size and euryhaline status associated with risk) is robust to whether vulnerable species are classified as threatened or non-threatened, but that the strength of the signal is sensitive to the classification scheme.

### Threats faced by species

The total number of species affected by each threat type showed that commercial harvesting is currently the primary extinction threat for marine ray-finned fishes, followed by pollution, development, and then climate change (Figure 4a), and we discuss each in order here. Our results are similar to what Miranda et al. (2022) have found for marine fishes using similar methodology and a

smaller random subset of marine fishes. Fisheries have long been recognized as exerting overharvesting pressure on fish populations, and many commercially fished species have depleted far below their historical baselines (Hutchings and Reynolds, 2004; McCauley et al., 2015; Arthington et al., 2016). The effects of ocean pollution on marine fish mortality and population trends are not as clear as the effects of harvesting. Oil spills can have negative impacts on fish stocks, especially given their threat of damage to embryos and larvae, but even the effects of oil spills remain understudied (Sørhus et al., 2015; Langangen et al., 2017). Pollution can also lead to lower species richness among fishes in a given ecosystem (Johnston and Roberts, 2009). However, while pollution is the second highest assigned threat type for these fish species, the overall effects of pollution on extinction risk and population declines for marine fishes are in greater need of study. Rapid human development, such as that occurring in the Persian Gulf and other regions around the world, can lead to a massive loss of suitable habitat, as coastal land is taken over for urbanization, industry, and shipping, driving declines in fish populations (Sheppard et al., 2010; Sale et al., 2011). The projected impacts of climate change on marine fishes are still a matter of ongoing research, but some results suggest climate change will have its biggest effect on tropical fishes and will be latitudinally selective in its impacts (Comte and Olden, 2017). Marine fishes as a whole may also be more threatened by climate change than terrestrial ectotherms, given that marine ectotherms operate at temperatures closer to their upper physiological limits (Pinsky et al., 2019). This could in turn drive higher rates of local extirpations relative to terrestrial ectotherms, as species seek out new locations within their thermal niches (Pinsky et al., 2019). We predict that climate change will affect a greater number of marine ray-finned fishes as climate change continues to increase and accelerate (IPCC, 2021).

We also found that threatened species are impacted by a greater number of threats, on average, than non-threatened species, in line with other studies (Figure 4b; González-Suárez and Revilla, 2014; Ducatez and Shine, 2017; Munstermann et al., 2022). Among terrestrial vertebrates, Munstermann et al. (2022) have found, on average, 2.66 threat types for threatened species and 2.37 threat types for non-threatened species. Likewise, we also found that threatened species (mean = 2.12, median = 2.00) face a greater number of threat types than non-threatened species (mean = 0.48, median = 0.00). Ducatez and Shine (2017) have found the number of threats assigned vary based on research effort, perhaps explaining, to some extent, why terrestrial species have more assigned threats on average, given the understudied nature of marine ray-finned fishes in comparison to terrestrial vertebrates. However, it also appears that the extent of human impact on the oceans is not yet as great as it is on land (McCauley et al., 2015). Because the IUCN does not require assigning threat types for species of least concern, it is possible that some threats are present but unrecorded for these species. Nonetheless, the threats for these species have not yet been severe enough to move them into a higher Red List extinction category. Collectively, these findings support that the "death by a thousand cuts" scenario (with threatened species on average facing a greater number of threat types than non-threatened species) is now a consistent signal across multiple vertebrate groups and that a multistressor perspective is necessary to accurately address the modern extinction crisis (González-Suárez and Revilla, 2014; Ducatez and Shine, 2017; Munstermann et al., 2022).

### Association of predictor variables with threat types

The association of body size with the four most common threat types may help explain why both larger and smaller fishes are at greater extinction risk. We found a positive association between increased body size and harvesting as a threat type, in line with prior studies (Olden et al., 2007; Genner et al., 2010; Figure 5a). In addition, we found that smaller fish species are more likely to have pollution as a threat type, suggesting exposure to pollution may be driving the heightened percentage of threatened smaller species (Figure 5b). Smaller fishes tend to have smaller home ranges (Kramer and Chapman, 1999; Luiz et al., 2013), and perhaps small home-range size is the reason we find smaller species to be more threatened by pollution and/or development – threats that may not simultaneously affect all regions of species with large home-ranges in the same way. Larger species may also be more motile and thus potentially better able to use more physically variable habitats and find spatial refuges from threats. In line with this reasoning, marine animals that are motile have had significantly lower extinction probabilities across the past 500 million years (Knope et al., 2020).

We also found that species that come into contact with brackish and freshwater are significantly more likely to have harvesting, pollution, and/or development as threat types (Figure 5). Pollution can adversely impact fishes that utilize freshwater environments, with some rivers receiving high levels of raw or undertreated sewage that create dead zones, serving as an impediment to fish migration (Limburg and Waldman, 2009). Dead zones from fertilizer runoff have also been increasing in the last 60 years, especially in coastal regions, and have affected at least 245,000 $km^2$ globally (Diaz and Rosenberg, 2008). Rivers have also been affected by other contaminants such as polychlorinated biphenyls (PCBs), which can potentially cause sublethal effects such as reduced fish survivability (Limburg and Waldman, 2009). The association of "marine, brackish, and freshwater" fishes with the development threat type may reflect the impact of dams. Dams are often associated with human development, and there are an estimated ~80,000 dams in the rivers of the United States alone, which impede the natural life history migrations of salmonids and other fishes (Jackson and Marmulla, 2001). Over 3,000 new dams are expected to be built worldwide as new renewable sources of energy are sought, potentially further increasing the threat posed by dams (Zarfl et al., 2015). Protecting diadromous fishes into the future could be helped by removing barriers like dams, installing fish passages when removal is not feasible, habitat restoration, restocking (breeding in hatcheries followed by release into the wild), and fisheries management for diadromous species commercially harvested (Verhelst et al., 2021). Dam removal could improve the outlook for diadromous fishes because habitat is found to rapidly improve when a dam is removed, which can help prevent local extirpation (Battle et al., 2016; Hill et al., 2019). Although we did not find a significant association with climate change, climate change is likely to alter hydrological patterns and raise water temperatures in many rivers, further threatening diadromous fishes (Schröter et al., 2005; Eliason et al., 2013). Forecasting future distributions of diadromous fishes under different climate change scenarios will be an important tool in mitigating this risk (Lassalle et al., 2008).

### IUCN considerations

Caution is warranted when interpreting the results for threat types, as the IUCN assigns threat types in a somewhat nonsystematic

manner. In contrast to the assignment to levels of extinction risk, which are determined using strictly quantitative criteria (IUCN Species Survival Commission, 2012), there is no consistent set of criteria used to determine whether a threat applies to a given species (Hayward, 2009; Cassini, 2011). Deficiencies can be present, such as highly threatened species with clearly defined threats not being assigned any threats (Hayward, 2009). Nonetheless, there is general agreement between IUCN threat types and other assessments of threat types, such as harvesting being the greatest threat for marine fishes, and the documented decline in commercially fished species (McCauley et al., 2015). Further improvement could be made by developing a more systematic and rigorous framework for assigning threat types, such as the one proposed by Cassini (2011), which calls for specialists to define objective values to measure threats and to assign threats by geographic regions.

Another concern with the IUCN is that the Red List may overestimate extinction risk under criteria A (population declines) for populations that undergo large population fluctuations due to harvesting. Overestimation of extinction risk has been documented in other taxonomic groups, such as corals, based on newly published abundance estimates (Dietzel et al., 2021), and overestimation of risk for commercially fished species has been a longstanding concern (Matsuda et al., 1998; Punt, 2000). However, Davies and Baum (2012) have found that the IUCN does not overestimate extinction risk in marine fishes and that there is strong alignment between fisheries reference points and IUCN status. Nonetheless, more consideration could be taken into account when evaluating commercially fished species given the expectation that populations will fluctuate in response to harvesting. A potential approach is the one called for by Miqueleiz et al. (2022), who suggest relying more heavily on criteria E (modeling of extinction risk) and incorporating management strategy evaluation, which utilizes simulations to judge the outcome of management actions (Punt et al., 2016), into assessments of the overall extinction risk of a species.

## Implications for ecosystems, evolution, and society

The sustained decline of these particularly at-risk larger species and species that move between marine, brackish, and freshwater environments may have disruptive impacts on the systems in which they reside or once were resident. For example, because larger-bodied species generally have larger home-ranges and biomass intake than their smaller-sized counterparts (McCauley et al., 2015), their loss may have an outsized impact on nutrient movement across ecosystem boundaries and generally reduce ecosystem connectivity (McCauley et al., 2015; Doughty et al., 2016; Ripple et al., 2017). These types of cross-system connections appear in some contexts to help stabilize the dynamics of these ecosystems (McCann et al., 2005; McCauley et al., 2018). Large species that move between marine and freshwater environments are similarly well known for hosting critically important connections between marine, coastal, and even terrestrial systems. Semelparous salmon, for example, that travel between marine and freshwater rivers release marine-derived nutrients when they die after breeding. Nutrients from these salmon carcasses affect many species of vertebrates and invertebrates and influence diverse processes from river biogeochemistry, in situ river productivity, growth rates of resident salmon, and riparian plant growth (Helfield and Naiman, 2001; Wipfli et al., 2003; Compton et al., 2006; Limburg and Waldman, 2009).

Large marine fishes can also often perform consequential roles as ecological engineers that are facilitated by their larger size (Moore, 2006). For example, the large size and bite force generated by the vulnerable humphead parrotfish (*Bolbometopon muricatum*), the largest of the parrotfish species, allows it to feed extensively on stony corals – a distinct niche from its smaller-bodied counterparts (McCauley et al., 2014). The direct and indirect trophic impacts of large at-risk fishes on other attributes of the food webs in which they are embedded can also be important, although highly context-dependent (Grubbs et al., 2016). Reductions in large predatory fishes driven by overfishing have been associated with increases in meso-predator and other prey abundances (Dulvy et al., 2004; Baum and Worm, 2009) as well as ecologically consequential changes in prey behavior (Madin et al., 2010).

Additionally, many of the fish species exhibiting elevated extinction risk represent deeply branching evolutionary histories, and their loss would result in disproportionate reductions in evolutionary history and diversity. For example, sturgeons are a deeply branching lineage and display remarkable capacity for morphological diversification despite their lack of species richness (Bemis et al., 1997; Rabosky et al., 2013), and are often considered to be the most threatened group of animals on the IUCN Red List (IUCN, 2010). Correspondingly, 17 of the 27 species of sturgeon in our dataset are listed as critically endangered. Many sturgeon species are also considered to be textbook examples of "death by a thousand cuts" scenarios, as many endangered populations have been negatively impacted by overharvest, dams, habitat degradation, and pollution (Bemis et al., 1997; Billard and Lecointre, 2000). Further declines and extinctions of sturgeon populations and species would therefore represent a substantial loss of unique fish evolutionary history.

The observed elevated vulnerability of larger fish and fish that move between marine, brackish, and freshwater environments may be similarly consequential for people and society. Many endangered larger fishes are (or were) commercially important. For example, the critically endangered beluga (*Huso huso*) and the endangered southern bluefin tuna (*Thunnus maccoyii*) represent extremely expensive seafood products whose full fishery value is not realized because of their current small population size and harvest restrictions (Gessner et al., 2010; Commission for the Conservation of Blue Tuna, 2020). Many fishes moving between marine, brackish, and freshwater environments are not only important to industry but are also of great importance to the culture and history of many coastal peoples, perhaps in part because of their relative ease of access for human harvesters, as well as their historical nutritional importance. Reductions in certain endangered salmonid species and populations provide illustrative examples of the sociocultural significance of declines of these fishes (Carothers et al., 2021).

## Conclusion

Given the current rates of population decline among marine ray-finned fishes (McCauley et al., 2015; Ceballos et al., 2017; IPBES, 2019), large-scale, systematic analyses are necessary to pinpoint which ecological traits are most strongly associated with extinction risk and which threat types are most strongly associated with these species. By leveraging the combination of recently available ecological, phylogenetic, and extinction risk profile data for a large proportion of marine ray-finned fishes, we demonstrate that species with larger body size and/or life histories that move between marine, brackish, and freshwater environments are more likely to be at elevated extinction risk. Further, we found that commercial

harvesting is currently the greatest threat to marine ray-finned fishes, followed by pollution, development, and then climate change, and that threatened species are simultaneously exposed to a greater number of threats, supporting a "death by a thousand cuts" scenario. This scenario now appears to be a common signal across both marine and terrestrial vertebrates (Munstermann et al., 2022). Harvesting is likely pushing larger fish and/or fish that move between marine, brackish, and freshwater environments toward greater extinction risk, while pollution may be pushing smaller fish and/or fish that move between marine, brackish, and freshwater environments toward greater extinction risk. The association of ecological traits with climate change is less clear at this point, and a more comprehensive systematic database of how species are threatened by climate change may help elucidate how climate change will impact species in the near future as the climate crisis continues to accelerate and intensify. Additionally, the determination of extinction risk status for data-deficient species (removed from this analysis) would help paint a more comprehensive picture of extinction risk. Altogether, our results demonstrate at a global level that larger fish and/or fish that move between marine, brackish, and freshwater environments are most at risk, and conservation effort could be directed toward more carefully managing these species. In particular, reducing pressures from overharvesting could help prevent further declines. Prioritization of conservation efforts to address these threats could help prevent global consequences for marine ecosystem structure and function and, in turn, human health and society.

**Open peer review.** To view the open peer review materials for this article, please visit http://doi.org/10.1017/ext.2023.23.

**Supplementary material.** The supplementary material for this article can be found at https://doi.org/10.1017/ext.2023.23.

**Acknowledgments.** We thank M. Munstermann for assistance with study methodology and R code. We thank A. Colon, L. Gavagan, S. Lewis, J. Regalario, C. Wine, and N. Zelaya for assistance with data collection. We thank J. Burns and B. Ostertag for assistance with manuscript edits and developing the research direction, and D. Koizuma for administrative support. We thank L. Revell for assistance with phylogenetic data sourcing and analysis. We thank the anonymous reviewers for their contributions. Any use of trade, firm, or product names is for descriptive purposes only and does not imply endorsement by the US government.

**Financial support.** This material is based on work supported by the US National Science Foundation under grant #1345247.

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
