## [Reviewer Report]

Comments on manuscript EXT-23-0007 “A Global Ecological Signal of Extinction Risk in Marine Ray-Finned Fishes”.

I enjoyed reading this paper. It is one of the first attempts to measure extinction risk in the large group of marine bony fishes (there is an interesting brand-new review on the drivers of modern marine extinctions: https://doi.org/10.1007/s10113-023-02081-8). Using GLM, the authors conclude that body size and euryhaline status, but not phylogenetic redundancy, are reliable predictors of extinction risk in marine fishes, with large and anadromous species being the most vulnerable. They also identified overexploitation and habitat alteration as the main threats to these highly vulnerable (bony) fish species.

I recommend this manuscript for publication after minor revisions. Here I present some comments that may add nuance to the Introduction, Discussion, and Conclusions sections.

Title: Perhaps using the term bony fishes in the title instead of ray-finned fishes could improve the bibliometrics of the article, as the word “ray” can be confused with the common name of the group of (sharks and) rays. The more technical category of ray-finned fishes can be used in the rest of the manuscript.

Lines 85-89: One-third of marine stocks are more or less overexploited. The available data support this particular assertion, but the overall picture is that the majority of fished populations are at sustainable levels of exploitation. Fisheries management should aim to increase this proportion at the expense of all other under and over-exploited stocks, but this does not necessarily mean restoring fish populations to pre-exploitation levels.

Lines 88-89: I would say “…some fish populations showing as much as 90% reduction…” unless there is sufficient evidence to clearly show that many marine fish species (including all their constituent populations) have been reduced to 10% of their carrying capacity.

Lines 95-100 (also lines 332-334): This paragraph gives me the message that marine fish extinctions are already happening. I’m aware that there are many studies suggesting significant declines in the numbers of several exploited marine fish populations, but the fact that there are no more than two or three documented fish extinctions (undoubtedly caused by anthropogenic impacts) cannot be ignored either. In the context presented by the authors, these two pieces of information would lead me to say that the potential for ecosystem-level impacts from a massive non-random extinction exists, but is far from being realized.

Line 145: In ecophysiological studies, body size, i.e. total length, is a common measure for comparing metabolic variables between population cohorts of a single species. However, when comparing between species, body mass, i.e. weight, is a surrogate variable that better represents the metabolic and ecological properties of the species (population growth, reproduction, recovery capacity, etc.). Perhaps an index combining length and weight would be ideal for describing aspects of a species that are relevant to the correlates of extinction. I don’t expect the authors to include body mass data in their analyses, but perhaps it is a relevant variable to consider in future marine fish extinction studies.

Line 144: Minimum population doubling time. One of the drawbacks of this variable is that it comes from a fishery model whose parameters (r, K and q) may not be constant in 1) highly variable fish populations, 2) abundance indices that show discernible environmental effects, and 3) abundance indices that show an increasing trend. In these cases, the statistical uncertainty associated with the parameter r (intrinsic growth rate) may be as large as the size of the categories used (less than 1.4, 1.4 - 4.4, etc.).

Line 145: similar to previous comment. Estimates of trophic level come from a variety of methods (i.e. stomach content, stable isotopes and ecopath models), each of which may have a large statistical uncertainty, which should not only be added or multiplied by the uncertainty of other predictors, but should also be reflected somewhere in the estimated extinction risk.

My final point concerns the relationship between IUCN categories (collapsed into threatened and non-threatened) and exploited fish populations. To achieve sustainable fisheries, the abundance of exploited stocks must be reduced by between 30% (for large, slow-growing fish species) and 70% (for small, fast-growing fish species) of their carrying capacity. This means that some fish species that are sustainably fished or slightly overfished could be included in the IUCN list simply because their abundance has been reduced by, say, 60%, which has little or nothing to do with their extinction.

A further level of complication is that an overexploited fish stock does not always mean that the entire species is threatened; a commercial fish species may comprise several populations and, although some of them may be overexploited, the risk of extinction for the species as a whole may increase only marginally. Finally, overexploitation is a gradient ranging from moderate to severe, but usually, no distinction is made between the two. Even in cases of severe depletion, the number of fish remaining in the sea is in the order of millions (e.g. Atlantic cod and bluefin tuna); a number small enough to threaten the viability of the fishery, but large enough to reduce concerns about extinction.

---

## [Reviewer Report]

There is no novelty in this study. Other similar studies use the same resources (Fishbase and IUCN Red List) and conclude the same. Several studies conclude that larger (marine) and diadromous fish species have the greatest extinction risk. Several studies analyse threats to marine fishes with similar results (because they analyse the same source, the red list). Sorry, but there is no novelty.

In general, I believe that Material and Methods need more explanation. I am not an expert on some statistical methods, such as the phylogenetics models, and I am unsure about this.

In the Fishbase database, you can check a vulnerability index (Cheung et al. 2005) proposed as a method to estimate vulnerability for marine fishes. This index includes data as the size and other variable. This index should be considered and discussed in this study.

Line 64-65: Take care because “species that move between marine and freshwater” are not the same as diadromous species. There are a lot of species that move between seas and rivers, but they are not migratory species.

Line 91: Please include a citation regarding the decline of the marine fish population.

Lines 102-119: Some of the proposed new analyses have been partially elaborated by Miranda et al. (2022), for example, the principal threats for bony fishes, and you should consider this study in your study.

Line 135: You do not need WoRMs to select marine bony fishes. Both Red List and Fishbase provide this information.

Line 144: Why only these five predictor variables? There are more useful variables in the Fishbase database. You have to explain that.

Line 146-148: What is the meaning of this sentence? You visualize this percent to include that on the analysis; how is this?

Line 168-172: Why? I do not understand why you categorise vulnerable species as non-threatened species. A vulnerable species is a threatened species. No doubt. Perhaps you do not have to analyse data in a binary manner but consider all the risk extinction categories. Your sensitive analysis (I do not know what this is; you must explain) should not be considered.

Lines 311-319: I am not sure about the “novel insights” of this study. Several studies conclude that larger (marine) and diadromous fish species have the greatest extinction risk. Several studies analyse threats to marine fishes with similar results (because they analyse the same source, the red list). Sorry, but there is no novelty.

Lines 500-522: In the same way, these conclusions have no novelty.

I believe that there are some papers that should be considered in the present study:

Cheung, W. W. L., Pitcher, T. J., & Pauly, D. (2005). A fuzzy logic expert system to estimate intrinsic extinction vulnerabilities of marine fishes to fishing. Biological Conservation, 124, 97–111.

Jelks, H. L., Walsh, S. J., Burkhead, N. M., Contreras-Balderas, S., Diaz-Pardo, E., Hendrickson, D. A., … & Warren Jr, M. L. (2008). Conservation status of imperiled North American freshwater and diadromous fishes. Fisheries, 33(8), 372-407.

Lassalle, G., Béguer, M., Beaulaton, L., & Rochard, E. (2008). Diadromous fish conservation plans need to consider global warming issues: An approach using biogeographical models. Biological conservation, 141(4), 1105-1118.

Miqueleiz I, Miranda R, Ariño AH, Ojea E. 2022. Conservation-Status Gaps for Marine Top-Fished Commercial Species. Fishes 7, 2.

Miranda R, Miqueleiz I, Darwall W, Sayer C, Dulvy NK, Carpenter KE, Polidoro B, Dewhurst-Richman N, Pollock C, Hilton-Taylor C, Freeman R, Collen B, Böhm M. 2022. Monitoring extinction risk and threats of the world’s fishes based on the Sampled Red List Index. Reviews in Fish Biology and Fisheries 32, 975–991.

Verhelst, P., Reubens, J., Buysse, D., Goethals, P., Van Wichelen, J., & Moens, T. (2021). Toward a roadmap for diadromous fish conservation: the Big Five considerations. Frontiers in Ecology and the Environment, 19(7), 396-403.

---

## [Editor Report]

This manuscript has received two disparate reviews. The first review was positive suggesting minor revisions, while the second suggested rejection for lack of novelty. The comments of reviewer #2 are perhaps less constructive, but they have a point that the novel aspects of this study need to highlighted in light of previous studies on the matter. A fuller appreciation of the previous literature (as suggested by reviewer #2) is thus mandatory. However, the extended dataset and the consideration of phylogenetic nestedness of traits are truly novel and deserve publication.

I agree with reviewer #1 that using bony fishes at least in the title will help avoid confusion. Please change the title accordingly. The link between exploitation, population size and extinction risk is worth emphasizing in the discussion given similar concerns in corals that the IUCN Red List may overestimate extinction risk (Dietzel et al. 2021, Nat Eco Evo, https://doi.org/10.1038/s41559-021-01393-4).

In sum, moderate revisions are required for this manuscript.

---

## [Reviewer Report]

The author’s replies are consistent, and yes, it is true; their arguments about the paper’s novelty are correct. The remaining comments about methodological, diadromous species, references and analysis were correctly replied.